# Pregnant women's and health workers' perceptions and experiences on the Rwandan ANC digital module intervention at selected health centres

Michael Habtu[1]*, Maria Barreix[2], Maurice Bucagu[1,3], Richard Kalisa[1], Nathalie Kayiramirwa Murindahabi[4], Fiacre Rugamba Rugero[5], Hedieh Mehrtash[2], Theopista J. Kabuteni[3], Tigest Tamrat[2], Rosemary K. Muliokela[2], Josiane Akingeneye[4], François Regis Cyiza[6], Uwimana Aline[6], Gilbert Uwayezu[7], Kama Mukamurigo Edith[1]

**1** University of Rwanda, College of Medicine and Health Sciences, School of Public Health, Kigali, Rwanda, **2** UNDP/UNFPA/UNICEF/WHO/World Bank Special Programme of Research Development and Research Training in Human Reproduction (HRP), Department of Sexual, Reproductive, Maternal, Child, Adolescent Health and Ageing, World Health Organization, Geneva, Switzerland, **3** World Health Organization, Kigali, Rwanda, **4** University of Rwanda, Single Project Implementation Unit (SPIU), Kigali, Rwanda, **5** University of Rwanda, centre of excellence in Biomedical engineering and e-health, Health informatics Department, Kigali, Rwanda, **6** Rwanda Biomedical Centre, Division of Maternal Child and Community Health, Kigali, Rwanda, **7** 1000Hills Solutions, Kigali, Rwanda

* mikel.habtu@gmail.com

## Abstract

As part of the New Antenatal Care Model for Africa and India (NAMAI) study, Rwanda implemented a digital module, in line with national digital health strategies, and the WHO SMART guideline framework. The purpose of this NAMAI study was to evaluate the acceptability and feasibility of implementing an updated national Antenatal Care (ANC) service package and the use of a digital tool to support and improve quality service provision. A qualitative component was conducted to explore the experiences of health workers and pregnant women on the implementation of the Rwandan digital ANC module intervention in study facilities. This qualitative study was conducted in 14 health centres in Nyanza and Nyagatare districts. A total of 13 heads of health centres and 14 nurses/midwives providing ANC services participated in Key Informant Interviews (KIIs). In addition, 10 Focus Group Discussions (FGDs) were conducted, each composed of seven to nine pregnant women. Data were collected in December 2024 using KII and FGD guides. All KIIs and FGDs were audio-recorded, transcribed verbatim and translated into English. Transcripts were analyzed employing using inductive thematic content analysis techniques with Atlas.ti Version 8. The Rwandan ANC digital module intervention was perceived to enhance tracking and follow up, improve data storage and reduce risk of record loss, simplify data analysis and reporting, and provide reminder notifications. However, some implementation challenges were highlighted, including slow performance of the digital tool, inadequate supervision, and increased workload due to the use of concurrent

**Data availability statement:** Publishing the transcripts from the qualitative component of our study would violate our approved protocol. Participants were not asked to provide consent for full-text data to be made publicly available, whether de-identified or not. Furthermore, in line with our protocol and WHO data-sharing policy, the study data are owned by the WHO member state and are not required to be publicly released. De-identified study data are archived within the WHO/HRP electronic record management system and are available upon request. Researchers interested in conducting secondary analyses with a specific research question may submit requests to srhmph@ who.int. More information about the study is available here: https://health-policy-systems.biomedcentral.com/articles/10.1186/s12961-023-01014-5.

**Funding:** This study is funded by the Bill and Melinda Gates Foundation (INV-001304 to MB), Sanofi Espoir Foundation, and the UNDP/UNFPA/UNICEF/WHO/World Bank Special Programme of Research, Development and Research Training in Human Reproduction (to MB). This content is solely the responsibility of the authors and does not necessarily represent the official views of the funders. The funders had no role in study design, data collection and analysis, decision to publish, or preparation of the manuscript.

**Competing interests:** The authors have declared that no competing interests exist.

paper and digital tools. Despite the perceived benefits of the Rwandan digital ANC module intervention, the study identified some challenges that may hinder its effective implementation. To optimize the delivery of ANC services through this digital tool and inform future scale-up, it is essential to address the mentioned challenges.

## Author summary

Digital health tools are increasingly used to improve the quality of maternal healthcare, but putting them into practice in low-resource setting remains difficult. Rwanda introduced a digital antenatal care (ANC) module aligned with the national digital health strategies and the WHO SMART guideline framework. This study explored how health workers and pregnant women experienced with the introduction of this digital tool in routine ANC services. We conducted interviews with heads of the health centres and nurses/midwives, as well as focus group discussions with pregnant women in 14 health facilities across two districts in Rwanda. Participants reported that the digital module improved the follow up of pregnant women, strengthened data storage, reducing the risk of losing records, simplified reporting and enabled timely reminders. At the same time, they highlighted key barriers, including slow system performance, limited supervision and the increased workload caused by using both paper and digital tools. Thus, strengthening system performance, reducing duplication of work and improving ongoing supervision could enhance service delivery and guide for future national scale-up.

## Background

Improving maternal and newborn health is a key global priority outlined in the Sustainable Development Goals (SDGs) agenda from 2016 to 2030 [1]. Further, the global agenda aims to reduce maternal mortality ratio to below 70 deaths per 100,000 livebirths and neonatal mortality to below 12 deaths per 1000 livebirths by 2030 [2,3]. Sub-Saharan Africa has the highest maternal mortality, accounting for 70% of global maternal deaths in 2023, [4] and the neonatal mortality rate in the region was estimated at 27 deaths per 1000 live births in 2022 [5]. Rwanda has significantly reduced the maternal mortality ratio from 1071 in 2000 to 149 per 100,000 livebirths in 2025 [6], and the neonatal mortality rate from 44 in 2000 to 19 per 1000 livebirths in 2020 [7], yet these rates remain higher than the global targets.

The reported maternal and neonatal deaths can be reduced significantly further, through provision of quality care services before, during and after childbirth [8,9]. Considering this, WHO released its first evidence-based guideline on ANC in 2016, which recommends eight contacts for all pregnant women rather than the four visits previously implemented [10]. The guideline contains 49 recommendations that cover

nutritional interventions, maternal and fetal assessments, preventive measures, management of common physiological symptoms and health system related interventions [11].

Since 2018, the governments of Rwanda, Zambia, Burkina Faso and two states from India (Assam & Tamil Nadu), with support from WHO, have adapted the 2016 ANC recommendations to their national settings to promote a positive pregnancy experience [12]. To understand what it would take to implement these updated national guidelines, the respective Ministry of Health (MoH) implemented the New Antenatal Care Model in Africa and India (NAMAI) study [12]. The purpose of the NAMAI study was to assess the acceptability and feasibility of implementing the updated national ANC service package to support and enhance maternal and newborn quality service delivery.

In addition to the updated national ANC package, the four countries implemented four co-interventions: i) training of health workers on the updated ANC package, ii) supervision on the updated ANC package, iii) provision of supplies and equipment and iv) community mobilization strategy. Rwanda and Zambia implemented a fifth co-intervention, country specific digital intervention to support the delivery of their ANC packages [13], in alignment with their national digital health strategies and clinical protocols and recommendations [14], and the WHO's SMART guideline framework [15]. As part of contextualization, the reference module was adapted using country-specific versions of the ANC Digital Adaptation Kit (DAK) [16]. The DAK provides software-neutral documentation that translates WHO recommendations into formats suitable for integration into digital systems [17].

Digital health interventions have gained increasing attention as scalable strategies to improve ANC utilization, quality care and pregnancy-related outcomes in sub-Saharan Africa. The interventions include a wide range of strategies from WhatsApp-based educational interventions to mobile applications, which improved pregnancy-related knowledge, reduced pregnancy-related anxiety, increased ANC contacts, skilled birth attendance and maternal engagement with health services [18–20]. Systematic reviews further indicate that women exposed to digital health intervention are significantly more likely to complete at least four ANC contacts, with some studies reporting improvements in skilled birth attendance and selected maternal and neonatal outcomes [21,22].

In Rwanda, as part of the formative phase of the NAMAI study, the WHO digital ANC module was adapted to the local ANC service package and setting; these are documented elsewhere [14,23]. The demonstration phase of the study then implemented the country-adapted Rwandan digital ANC module to support the MoH's longer-term goal of scaling up digital interventions. The key features of the digital module include: offline access with synchronization; client registration; clinical decision-support algorithm for routine ANC documentation such as history, physical exam, symptom tracking, diagnostics, counseling or treatment; integration with national reporting systems such as DHIS2; SMS reminders to pregnant women; filter specific cases easily; and identify women who are overdue for ANC contacts [14,23].

Although digital interventions have demonstrated potential to improve service utilization and maternal and neonatal outcomes in sub-Saharan Africa [18–20], evidence remains limited regarding user experiences, workflow integration, and real-world implementation. Thus, this study aimed to deploy qualitative methods to explore the experiences and perceptions of health workers and pregnant women regarding the Rwanda digital ANC module intervention in selected health centres.

## Methods

NAMAI study implementation was carried out through a collaborative effort led by the MOH/Rwanda Biomedical Centre, University of Rwanda as the implementation partner, digital intervention leads 1000Hills Solutions, and technical support from the WHO.

### Ethics statement

The study received approval from the Rwanda National Ethics Committee (Ref No: 650/RNEC/2021) and the WHO Ethics Review Committee (ID#: A66008C), with annual evaluation of the study progress. A written informed consent form was

provided in Kinyarwanda (local language) for each participant prior to the interviews and/or discussions. Written or audio records were de-identified. Confidentiality and privacy were strictly maintained throughout the study. We conducted the FGDs and KIIs in private rooms to prevent unauthorized access or interruptions, and all consent forms and related documentation were securely stored in a locked location to ensure that participant information remained confidential.

### Study design and setting

This qualitative phenomenological study is part of the NAMAI implementation research study [12]. The study was carried out in the Eastern Province's Nyagatare district and the Southern Province's Nyanza district. The intervention was implemented in 14 health centres, 7 from Nyanza and 7 from Nyagatare district. The digital module was implemented from March 2024 to October 2024. We included health centres in the study if they maintained a minimum monthly volume of 30 ANC attendees, possessed at least two health workers providing ANC, featured a dedicated ANC service delivery room, and their management agreed to participate.

### Digital intervention

The Rwandan digital module was implemented by trained and supervised health workers, following extensive consultations with pertinent stakeholders and informed by qualitative research among health workers to ensure its feasibility, acceptability and contextual relevance. Health workers, including ANC service providers, data managers and heads of health centres participated in a three-day training on how to use the Rwandan ANC module. Following the training, each health worker providing ANC services was provided a tablet for implementing the tool during ANC service delivery. In addition, monthly supervision and monitoring was conducted by the local technology partner (1000Hills Solutions team). The details of the digital intervention is described in [12,23].

### Participants

We recruited a total of 106 participants, comprising 13 heads of health centres (HCs), 14 nurses/midwives in providing ANC, and 79 pregnant women. Pregnant women participated in 10 FGDs, each consisting of seven to nine participants. All the HCs that were implementing the NAMAI study were included. Within these health centres, heads of HCs and nurses/midwives providing ANC services, participated in the study. Pregnant women were purposively recruited from the selected HCs on the basis of having attended at least two ANC contact including the current visit. They were recruited from the selected facilities during the ANC sessions, and group discussion was conducted after they had all received the services.

### Instruments and procedure

Data collection for this study employed both KII and FGD guides. We conducted FGDs with pregnant women, while KIIs were held with heads of HCs and nurses/midwives providing ANC services. The tools for discussion and interview guides were initially developed in English, then adapted to the Rwandan context. The FGD guide for pregnant women (S1 Text), KII guide for heads of health centres (S2 Text) and KII guide for nurses/midwives providing ANC services (S3 Text) were specifically developed for this study and tailored to ensure contextual relevance and appropriateness for capturing participants' perspectives. In addition, to ensure clarity and cultural relevance, we translated the guides into local Kinyarwanda language.

We conducted the FGDs and KIIs in a quiet and private setting within the selected HCs to facilitate open and uninterrupted conversation. During group discussion, the pregnant women were seated in a circular manner to promote inclusivity and active participation. The FGD with pregnant women lasted about 60 minutes while the one-on-one KIIs with the health workers lasted approximately 45 minutes.

We trained four research assistants with prior experience in qualitative research. We audio-recorded all FGDs and KIIs using digital devices to ensure the accurate capture of participants' perspectives or experiences. In addition to the audio-recordings, the research assistants took comprehensive field notes to document contextual observations. While guides were used to direct the interviews and discussions, when necessary, probing questions were used to elicit deeper insights from participants. Throughout the data collection period, the research team provided close supervision to ensure the accuracy, completeness and overall quality of the data collection.

## Data analysis

Qualitative data obtained from the KIIs and FGDs were transcribed verbatim in Kinyarwanda and subsequently translated into English. The translated transcripts were reviewed and verified by FRR, JK and JM, all whom are proficient in both English and Kinyarwanda. Following verification, the transcripts were imported into Atlas.ti. Version 8 [24], for organization and initial coding. As outlined by Braun and Clarke [25], an inductive thematic analysis approach was conducted using the six phases. These include familiarization of data, generation of codes, combining codes into themes, reviewing themes, determining the significance of themes, and reporting of findings. To ensure credibility and validity of data analysis, two research team members (MH and JM) independently coded the data. They subsequently convened regular meetings to compare, discuss, and reconcile any discrepancies in codding decisions.

## Results

### Socio-demographic characteristics of the pregnant women

The average age of the pregnant women participated in the FGDs was 28.8 years, with approximately one-third were in the 19–24 years age group (Table 1). Most of the pregnant women (67.1%) had attained primary level of education. The majority were cohabiting (57.0%) and on average, the women had 3 pregnancies. Approximately one-third of women (30.4%) attended four ANC contacts.

### Socio-demographic characteristics of the health workers

Most of the health workers (48.2%) were aged between 35 and 44 years, with an average age of 39.44 years. The majority held a diploma (74.1%) and 12 had more than six years of experience (44.4%) (Table 2).

### Qualitative findings from interviews and group discussions

Two overarching themes emerged following thematic analysis including perceived benefits of digital module intervention and barriers to its successful implementation. The detailed sub-themes are shown in Fig 1.

### Perceived benefits of the Rwandan ANC digital module intervention

**Perceived enhanced tracking and follow up.** Most of the heads of HCs and nurses/midwives providing ANC services emphasized that the digital module has greatly improved follow-up processes. It has made it easier to track pregnant women who miss their scheduled appointments and to make necessary referrals. One of the key benefits they highlighted was the tool's ability to provide real-time information on pregnant women who have been lost to follow up. This helped to identify those who missed appointments and take quick action leading to improved retention and continuity of care.

*The digital tool has improved follow-up. It allows us to track pregnant women who miss appointments and contact them for rescheduling. This has helped reduce missed contacts. The digital tool also reminds what was done previously and helps plan the next steps.* (**P-11, Head of HC**)

**Table 1. Socio-demographic characteristics of the pregnant women (N = 79).**

| Attributes | N (%) |
| --- | --- |
| **Age [years]** | |
| 19–24 | 26(32.9) |
| 25–29 | 19(24.1) |
| 30–34 | 12(15.2) |
| 35 and above | 22(27.8) |
| Average age (+SD) = 28.8(6.7) | |
| **Level of education** | |
| None | 1(1.3) |
| Primary | 53(67.1) |
| Secondary | 25(31.6) |
| **Marital status** | |
| Married | 27(34.2) |
| Cohobating | 45(57.0) |
| Separated | 7(8.9) |
| **Number of pregnancies** | |
| One | 19(24.1) |
| Two | 21(26.6) |
| Three | 19(24.1) |
| Four and more | 20(25.3) |
| Average age (+SD) = 2.7(1.4) | |
| **Number of ANC visits at the Facility** | |
| Two | 20(25.3) |
| Three | 17(21.5) |
| Four | 18(22.8) |
| Five and more | 24(30.4) |
| Average age (+SD) = 3.7(1.6) | |

**Table 2. Socio-demographic characteristics of the health workers (N = 27).**

| Attributes | N(%) |
| --- | --- |
| **Age** | |
| 25–34 | 8(26.6) |
| 35–44 | 13(48.2) |
| 45 and above | 6(22.2) |
| Average age (+SD) = 39.44(8.35) | |
| **Level of education** | |
| Bachelor | 7(25.9) |
| Diploma | 20(74.1) |
| **Length of time working at current position** | |
| 1 to 3 years | 9(33.3) |
| 4 to 6 years | 6(22.2) |
| 7 years and above | 12(44.4) |
| Average duration of stay (+SD) = 7.26(5.93) | |

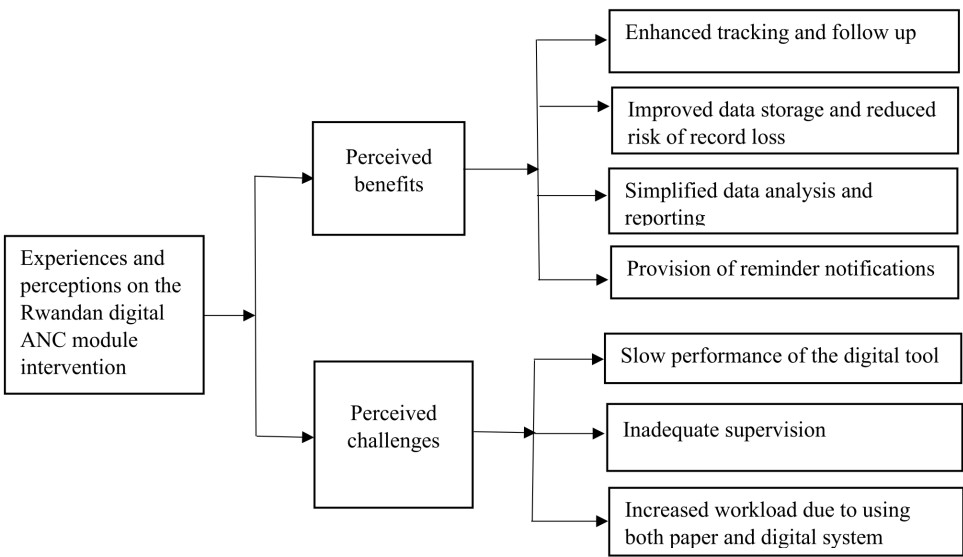

**Fig 1. Core themes and sub-themes.**

*This is a good tool; it has helped us a lot because it keeps all the information of the pregnant woman. Like when a pregnant woman comes for a second contact then you can immediately retrieve all her previous information without going to look in the forms or registers.* **(P-2, a nurse/midwife providing ANC)**

*The digital tool is important to closely monitoring each pregnant woman, once you learn to use the digital tool, you immediately notice if a pregnant woman has any danger signs. Through the paper register, you might miss those red flags on danger signs, but in the digital tool, you can go back and it clearly shows that this woman needs referral because she has these signs.* **(P-14, a nurse/midwife providing ANC)**

Similarly, the pregnant women reported that when they inquire about their next appointments and number of contacts, the health workers were immediately able to retrieve the information using the digital tool.

*When you accidently lose the appointment card, it was a very serious problem to get another one before, but now they look in the tool using the name and offer a new appointment easily.* **(FGD-2)**

*The difference that I observed in using the smartphone [digital module] is that when they registered you, they retrieve your information when you come for the next contact without asking you again. In addition, when there is a place where you may have forgotten, they check in the tool and remind you about how the situation was.* **(FGD-4)**

**Perceived improved data storage and reduced risk of record loss.** Health workers highlighted that the implementation of digital tool has not only improved the storage of maternal health but also substantially reduced risk of losing records. They described that the tool provides a secure and reliable storage for antenatal records, ensuring that the information remains readily accessible for future use.

*The digital module has been useful in recording pregnant women's information securely, reducing the risk of losing records.* **(P-2, Head of HC)**

*The information of the pregnant women is stored in a safe way, so that the information will be available in the coming years.* **(P-11, Head of HC)**

Pregnant women in the focus group discussions confirmed that before the introduction of the digital tool, information collected during the initial antenatal contacts were frequently misplaced or lost, compromising the ability to provide consistent and appropriate care at subsequent contacts.

*Before using of this technology, information provided during the first visit could be lost, making it difficult to provide appropriate care during subsequent visits. The introduction of the digital tool has prevented such loss of information.* (**FGD-3**)

**Simplified data analysis and reporting.** Another advantage of the digital tool was its ability and capacity to analysis and prepare routine reports. It was indicated that the tool performs basic analysis and generates summary statistics about the status of ANC services. The health workers also contrasted this with the precious paper-based system, which required manual counting and calculations to compare reports. Moreover, it facilitates the preparation of monthly reports at ease.

*The good side of this tool is that it is capable of doing simple analysis providing summary statistics that describes the functioning status of the ANC services. In addition, it assists in doing the monthly report for MoH.* (**P-5, Head of HC**)

*In the past, we used paper-based systems, which required manual counting and took a lot of time. Now, the digital model allows us to generate reports quickly and accurately, with no errors.* (**P-13, Head of HC**)

**Provision of reminder notifications.** One of the key benefits of the digital tool indicated by the health workers was sending reminder messages via phone to pregnant women about their upcoming scheduled appointments, which has resulted reduced number of pregnant women missing appointments.

*The digital tool sends a text message to pregnant women, reminding them of the date of appointment. This is especially helpful because pregnant women are not necessarily ill, so they may forget their appointments. The SMS system ensures they are reminded and show up on time.* **(P-13, Head of HC)**

*The digital tool reminds pregnant women by giving them messages on their phones. Due to this we have observed that there has been reduced the number of women who miss their appointments.* (**P-1, a nurse/midwife providing ANC**)

In the group discussion among the pregnant women, it was stated that the notification they receive regarding the appointment was crucial as expressed in the following quote.

*While registering us, the health provider asked our phone numbers, this is also helpful because we receive messages on phones reminding about the scheduled appointment time.* (**FGD-4**)

However, it was noted that there were some women who do not have phones or change their numbers which posed a barrier to receiving the notification. In addition, some women reported not receiving SMS reminders on their phone which requires further investigation to identify the issue.

*The challenge is if a pregnant woman does not have a phone or changes her number, it becomes difficult to reach her. In such cases, we rely on community health workers to follow up.* (**P-3, Head of HC**)

*The message sent to pregnant women was very helpful in reminding about their appointment, but there was a problem that there were few who did not receive the message.* (**P-4, a nurse/midwife providing ANC**)

## Barriers for the Rwandan ANC digital module intervention

**Slow performance of the digital tool.**  The health workers expressed that slow performance of the digital tool was a key challenge. They noted that delays in its speed disrupt service delivery, making it difficult to provide ANC services and increasing the time required for consultations. Although, they reported being familiar with the tool following training, they emphasized that its slow operability remains a major challenge. Some also reported that, due to its slowness during the updating, they record information manually and later enter into the digital tool after work.

*After the training, I became familiar with it, but the main challenge remains its slowness and takes a lot of time. Apart from that, I understood how to use it easily.* (**P-14, a nurse/midwife providing ANC**).

*The digital tool works slowly, and things take long time. If there was a way to make it faster moving from one page to another, it would be better. This would prevent spending too much time.* (**P-12, a nurse/midwife providing ANC**).

*The slow system makes it difficult to update records. Sometimes, I have to first record information in the book and then later update it in the digital system, which is time-consuming.* (**P-3, a nurse/midwife providing ANC**).

Further the ANC providers recommended strengthening the digital tool by drawing lessons from other systems and implement a stronger version that makes it work faster.

*The developers could take lessons from other systems like the cancer registry and the vaccination tracking system to improve the speed of this one. At the facility, we already have a lot of work. So, when the system runs slowly, it becomes frustrating and feels like doing double work.* (**P-14, a nurse/midwife providing ANC**).

*My advice is that this system should be strengthened and make it work faster.* (**P-1, a nurse/midwife providing ANC**).

In addition to the slow performance of the digital tool, poor internet connectivity was frequently cited as a major challenge. Although, the tool was designed to function in offline mode, internet connection was still essential for signing in, ensuring proper folder synchronization and enabling data transfer once data collection had been completed offline.

*The tool requires a good internet connection when starting the digital tool, which is sometimes unreliable. This can hinder the recording process and delay the services.* (**P-2, Head of HC**)

*Here where we work, the internet connection is slow, and sometimes when we notice it is not working well, we carry the forms with us and complete them at home, where the connection is better and more reliable*. (**P-11, a nurse/midwife providing ANC**)

*At the beginning of using this system was very difficult and challenging for us: you enter a patient information, and then the system stops due to weak connectivity.* (**P-5, a nurse/midwife providing ANC**).

**Inadequate supervision.**  It was noted that the initial training on the digital tool was generally adequate, however, the subsequent supervision was insufficient. The follow up from supervisors were inadequate, with some facilities receiving only a single supervisory visit since the tool's introduction.

*Supervision for the digital model were not sufficient. More follow-ups are needed to ensure all health workers can use the tool effectively.* (**P-5, Head of HC**)

   

*We only had supervision once, there was no other supervision. My suggestion in order to use digital tool well is to increase supervision from project management, district hospital and Rwanda Bio-medical Centre.* **(P-1, Head of HC)**

*We were well trained and we appreciate those who training us, but I think we did not have enough supervision.* **(P-3, Head of HC)**

**Increased workload due to using both paper and digital tools.** Health workers expressed significant concerns about the increased workload resulting from the simultaneous use of both paper based and digital tools. They indicated that, despite the introduction of the digital tool, existing paper-based such as registers which require manual documentation, remain in use. Consequently, the digital tool has added extra responsibilities and work.

*One challenge is the dual use of both digital and paper-based systems, which increases workload. Additionally, data management remains a challenge, as we have not yet trained dedicated data managers for the digital tool* **(P-7, Head of HC)**

*We have adapted ourselves to the digital tool and we found it very useful and helpful, but it increased the workload as the usual system of writing in papers is still there.* **(P-10, a nurse/midwife providing ANC)**

The increased workload was also associated with increased time resulted long waiting times for pregnant women.

*It is time-consuming, due to the need for entering data into the digital tool and paper-based records.* **(P-2, Head of HC)**

*It would be better if we use only the digital tool and stop using paper files, which slow things down.* **(P-13, a nurse/midwife providing ANC)**

## Discussion

Our study revealed that perceived benefits of the digital intervention including enhanced tracking and follow up, improved data storage and reduced risk of record loss, simplified data analysis and report, as well as provision of reminder notifications. However, some challenges that limited utilization of the digital module at the point of care were identified such as slow performance of the tool, inadequate supervision, and increased workload due to using both paper and digital tools.

The health workers in our study perceived that the digital tool has considerably improved follow-up by simplifying the tracking of missed appointments and improving appointment scheduling. This is aligned with an evaluation study conducted in Kenya, which showed that a mobile health tool helped community health workers (CHWs) track vital events more efficiently compared to the paper based tracking system during pregnancy [26]. Similarly, a pre/post intervention study in Nigeria demonstrated CHWs were able to follow-up women on scheduled contacts using mobile application [27]. It is evidenced that digital tools in low and middle income settings have promising effect on better quality care [28,29], as well as improved monitoring of patients treatment adherence [30,31]. Thus, the Rwandan digital ANC module might have helped to identify those who missed appointments and might have facilitated prompt clinical action leading to improved retention and continuity of care.

The present study further highlighted that the digital tool enhances data storage, reduces risk of record loss and facilitates the long-term usability of information compared to paper-based system. Similar studies indicated that mobile app-based electronic health record improves convenience of data storage, sustainability of data and less likely to be lost [32–35]. Our study also found that the digital tool is able to generate report that simplifies analysis and documentation, as reported in other studies [27,32,36,37]. These findings reflect the ability of the digital tools to effectively store, manage data and simplify in generating reports in a real time manner. However, it should be noted that studies used varied digital component interventions and design which makes it difficult for direct comparison.

Consistent with other studies [32,38,39], our study highlighted that one major benefit of the digital module is its ability to send phone reminders for scheduled appointments. These reminders were perceived as effective mechanisms for improving attendance and adherence to the ANC contacts. However, some women did not receive the notification messages despite having provided their phone numbers, a situation that warrants further investigation. However, consultation with the technical team revealed a configuration issue in the digital system that was preventing SMS reminders from being sent to pregnant women. After a data migration, all women began receiving SMS reminders. A limitation noted was that some women do not own mobile phones, making it difficult for them to receive notification messages.

In terms of challenges, slow performance or speed of the digital tool was a significant barrier as cited by the health workers in this study. Similar results have been documented in other studies, which reported that slow performance is a significant barrier to the effective use of digital technologies [40–42]. It was noted that health workers use paper record when the tool is down then transfer data form paper records into digital tool, which might introduce errors [43]. Moreover, slow performance of digital tools can interrupt workflow processes and negatively affect patient care by delaying timely access to critical patient important information [40]. Alongside the tool's slow performance, limited internet connectively for singing, synchronization and updating data emerged as a challenge, aligned with findings reported in other studies [44,45]. Although designed for offline use, the tool nonetheless required an internet connection to enable proper synchronization of folders, which at times making difficult, to retrieve some records. This finding underscores the importance of optimizing robust offline functionality to ensure continuity of use in contexts with unstable internet connectivity.

The nurse/midwife providing ANC and the heads of health centres highlighted that the ongoing supervision and follow-up support were insufficient to ensure optimal utilization of the digital tool. This finding is in agreement with other studies [46,47]. This might have resulted limited opportuneness for onsite support and guidance, thereby constraining the full and effective use of the digital tool. Regular supervision programs are essential to build user confidence and competence, thereby facilitating successful use of digital tools into routine clinical practice [48]. This underscores the need for strengthened supervisory structures and sustained capacity building initiatives to optimize the implementation and effective used of the digital tools in maternal service care settings.

Our findings showed simultaneous use of both paper-based and acceptance of digital tool utilization despite its associated increased workload when used both. This was consistent with several studies. Dual data entry using digital tool and paper-based records is a key barrier in implementing digital tool, which is considered as double work [32,40,49–52]. This might lead to poor data entry and compromised service quality, particularly when there many clients, as claimed by some of the study participants. Continued reliance on paper-based documentation often reflects existing policy, reporting requirements, and regulatory mandates, which may slow the transition to fully digital workflows. This highlights the need to strengthen efforts in transitioning from paper-based systems to digital tools in order to reduce duplication of work.

Collectively, the barriers identified in this study have important implications for national policy and the scale-up of the digital ANC module in Rwanda. Addressing these challenges requires the establishment of national digital health standards for system performance, policies that prioritize offline functionality to support equitable implementation in low-connectivity and rural settings, and formal integration of digital health governance and oversight within existing supportive supervision systems. Harmonization of the digital ANC module with national HMIS/DHIS2, the national Electronic Medical Records (EMR) system e-Buzima, and reporting requirements is essential to ensure interoperability and reduce duplication. Moreover, the shift from demonstration projects to routine national implementation requires proactive planning for workload management, clear institutional roles, and sustained engagement of key stakeholders, including the MOH and RBC.

## Strengths and limitations

This study is the first in a rural Rwandan setting to examine health workers' and pregnant women's views and experiences with the digital ANC module, with adequate sample size can be considered a strength of the study. Triangulation from the

different study groups involved in key informants and group discussions was also another strength of the study. However, one of the key limitations of this study is the exclusion of certain stakeholders, particularly representative from the Ministry of Health and supervisory personnel. Their perspectives could have provided valuable insights into the broader context of digital tool implementation and nationwide scale-up.

## Conclusion

Our study showed perceived positive benefits of the Rwandan digital ANC module intervention including improved tracking and follow up monitoring, secure data storage to prevent record loss, simplified data analysis and reporting, and the provision of reminder notifications. Despite these benefits, the study provided in-depth insights about the barriers hindering the effective implementation of the digital tool. Among the most significant barriers were slow performance, inadequate supervision as well as increased workload and time due to using both paper and digital tools. Therefore, for further improved care services of the digital tool, the challenges highlighted in our study should be considered. Moreover, in order to investigate the effect of the digital tool during ANC services on pregnancy and childbirth related outcome, further research is needed.

## Supporting information

**S1 Text. Focus Group discussion guide for pregnant women regarding the adapted Rwanda ANC Digital Module.**
(DOCX)

**S2 Text. Key informant interview guide for heads of health centers regarding the Rwandan ANC Digital Module.**
(DOCX)

**S3 Text. Key informant interview guide for nurses providing ANC services regarding the Rwandan ANC Digital Module.**
(DOCX)

## Author contributions

**Conceptualization:** Michael Habtu, Maria Barreix, Kama Mukamurigo Edith.

**Data curation:** Michael Habtu, Nathalie Kayiramirwa Murindahabi, Fiacre Rugamba Rugero, Josiane Akingeneye, Kama Mukamurigo Edith.

**Formal analysis:** Michael Habtu, Kama Mukamurigo Edith.

**Funding acquisition:** Maria Barreix.

**Investigation:** Michael Habtu, Maria Barreix, Kama Mukamurigo Edith.

**Methodology:** Michael Habtu, Maria Barreix, Kama Mukamurigo Edith.

**Project administration:** Kama Mukamurigo Edith.

**Resources:** Kama Mukamurigo Edith.

**Software:** Michael Habtu, Kama Mukamurigo Edith.

**Supervision:** Michael Habtu, Kama Mukamurigo Edith.

**Validation:** Michael Habtu.

**Visualization:** Michael Habtu.

**Writing – original draft:** Michael Habtu.

**Writing – review & editing:** Maria Barreix, Maurice Bucagu, Richard Kalisa, Nathalie Kayiramirwa Murindahabi, Hedieh Mehrtash, Theopista J Kabuteni, Tigest Tamrat, Rosemary K Muliokela, François Regis Cyiza, Uwimana Aline, Gilbert Uwayezu, Kama Mukamurigo Edith.

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
