## [Decision Letter · Decision Letter 0]

29 Dec 2025

Response to Reviewers
Revised Manuscript with Track Changes
Manuscript
**Journal Requirements:**

i. Please clarify all sources of financial support for your study. List the grants, grant numbers, and organizations that funded your study, including funding received from your institution. Please note that suppliers of material support, including research materials, should be recognized in the Acknowledgements section rather than in the Financial Disclosure.

ii. State the initials, alongside each funding source, of each author to receive each grant. For example: "This work was supported by the National Institutes of Health (####### to AM; ###### to CJ) and the National Science Foundation (###### to AM)."

iii. State what role the funders took in the study. If the funders had no role in your study, please state: “The funders had no role in study design, data collection and analysis, decision to publish, or preparation of the manuscript.”

iv. If any authors received a salary from any of your funders, please state which authors and which funders.

2. Please ensure that your Ethics Statement is available in its entirety at the beginning of your Methods section, under a subheading 'Ethics Statement'.

3. Please provide separate figure files in .tif or .eps format.

4. We note that you have included your Figures within the body of your manuscript. Please remove the Figures from the body of your manuscript and upload them as separate Figure files.

5. In the online submission form, you indicated that “The data used in this study are not publicly available due to the sensitivity of the qualitative data. De-identified data may be available upon request; please contact the corresponding author.”.

3. Uploaded as supplementary information.

**Additional Editor Comments (if provided):**

Thank you for submitting your manuscript, “Pregnant women’s and health workers’ perceptions and experiences on the Rwandan ANC digital module intervention at selected health centres,” to PLOS Digital Health.

After careful review, the reviewers agree that this is a well-conducted qualitative study addressing an important and timely digital maternal health intervention. The study design is appropriate, the analysis is methodologically sound, and the conclusions are supported by the data. Reviewers highlighted the strengths of the qualitative approach, the multicenter design, and the relevance of the findings for digital ANC implementation in low-resource settings. Only minor revisions are required to strengthen clarity, positioning within the digital health literature, and presentation.

Please address the following points in your revision. These reflect the full set of reviewer comments, consolidated and organized for clarity.

1. Positioning within the digital health literature

Reviewers noted that the Background and Discussion would benefit from more clearly situating this study within the broader digital health and maternal health literature, particularly in sub-Saharan Africa. Please:

Include a paragraph contextualizing the role of digital health tools in improving ANC uptake and quality in sub-Saharan Africa.

Clarify the specific evidence gap this study addresses.

Distinguish what this paper adds beyond prior NAMAI- or SMART-related publications, particularly in terms of user experience, workflow integration, or real-world implementation learning.

2. Clarifying the digital health contribution and interpretation

Several findings align with broader digital health implementation literature. Reviewers recommend:

More explicitly framing the results as digital health design and implementation lessons rather than effectiveness evidence.

Tightening language in a few places to clearly distinguish perceived improvements from objectively measured outcomes.

Including a brief synthesis in the Discussion that highlights implications for system speed, offline functionality, supervision models, and workflow integration.

3. System integration, scale-up, and implementation considerations

Reviewers raised several related points that should be addressed with modest expansion:

Strengthen discussion of DHIS2 integration, supervision, and continued paper–digital dual use.

Add brief reflections on how policy, reporting requirements, or regulatory factors may contribute to continued paper-based documentation.

Clarify implications for transitioning from demonstration projects to routine national implementation, including workload considerations and stakeholder involvement (e.g., Ministry of Health).

Explain why some participants did not receive text messages and whether this has implications for nationwide rollout.

Where relevant, comment on transition strategies for increased workload and potential cost considerations.

4. Methods clarification and analytic transparency

To improve clarity and replicability, please:

Clarify whether the thematic analysis followed an inductive or theory-driven approach within the Braun and Clarke framework.

Consider making interview and focus group discussion guides available as supplementary materials, if feasible.

5. Editorial, language, and data reporting issues

Reviewers identified several minor but important presentation issues:

Address grammatical and typographical errors throughout the manuscript (e.g., “algorism” should be “algorithm”).

Ensure consistent statistical terminology in Tables 1 and 2 (e.g., “Mean” rather than “Average”).

Proof-read all sections to improve clarity and readability.

Confirm that the Data Availability Statement fully complies with PLOS policy and clarify any inconsistencies noted by reviewers.

Please provide a point-by-point response to the reviewer comments, indicating where changes have been made or briefly explaining when a suggestion was not adopted.

I look forward to receiving your revised manuscript.

Kind regards,

Gloria Aidoo-Frimpong, PhD, MPH, MA

Academic Editor

**Reviewers' Comments:**

**Comments to the Author**

1. Does this manuscript meet PLOS Digital Health’s publication criteria?

Reviewer #1: Yes

Reviewer #2: Partly

Reviewer #3: Yes

2. Has the statistical analysis been performed appropriately and rigorously?

Reviewer #1: Yes

Reviewer #2: Yes

Reviewer #3: Yes

3. Have the authors made all data underlying the findings in their manuscript fully available (please refer to the Data Availability Statement at the start of the manuscript PDF file)?

Reviewer #1: No

Reviewer #2: Yes

Reviewer #3: Yes

4. Is the manuscript presented in an intelligible fashion and written in standard English?

Reviewer #1: No

Reviewer #2: No

Reviewer #3: Yes

Reviewer #1: Overall:

The authors present an evaluation of health workers’ and pregnant women’s perceptions and experiences of the Rwandan ANC digital module. The study design and data analysis methods are relevant to the objectives. The authors provide a balanced view in their results highlighting both benefits and challenges of the ANC digital module and the conclusion reflects their findings. The manuscript is well structured and easy to read through. However, the Background section can be further improved by contextualising the role of digital health and its implementation in sub-Saharan Africa and highlighting the evidence gap that this research fills. The manuscript has some typographical errors in all sections and will benefit from proof-reading and correction.

Background:

The Background introduces maternal and neonatal mortality statistics and highlights the policy change to 8 ANC contacts as a strategy to address this. The authors then proceed to describe the NAMAI study narrowing down to Rwanda where the digital ANC module was implemented.

Major comment: The focus of the paper is on the ANC digital module intervention. The Background section should situate the role of digital health tools in promoting healthcare/ANC uptake (in sub-Saharan Africa). I recommend including a paragraph on this and clarifying the knowledge gap that the research is looking to fill.

Minor comments:

1. Lines 67-75 (the first paragraph) presents a lot of statistics including multiple regions (sub-Saharan Africa, Central and Southern Asia). The authors can make this more succinct and present only the most important figures to make their point.

2. Line 77-78: Reference [8] is used as evidence to support the statement “maternal and neonatal deaths can be reduced significantly further, through provision of quality care services before, during and after childbirth” – yet the reference is to a protocol. I suggest using alternative references with evidence to support this.

Methods:

The authors explain the site selection, participant selection, data collection and data analysis processes.

Minor comments:

1. The Methods section has generally been written in passive voice. I suggest the authors consider using active voice to minimize ambiguity as to who was performing the different tasks.

2. Line 169: The authors state that they followed the Braun and Clarke thematic analysis process. Can they clarify whether they conducted Inductive analysis or Theoretic analysis of the data.

3. Will the interview/FGD guides be made available in the supplementary materials? This can help in understanding the responses and replicating the research elsewhere.

Results

The authors present the sociodemographic characteristics of the pregnant women and health workers in prose and 2 tables that have clear headings. They present the interview and group discussion findings categorised in themes and sub-themes. No further comments.

Discussion and Conclusions

They fit with the aims of the study and reflect the result they obtained. They have compared the findings with those of other studies, highlighted the key limitations and provided recommendations in line with their findings. No further comments.

Reviewer #2: To further strengthen the manuscript, the authors may consider the following refinements:

Clarify the digital health learning: Many of the findings resonate with broader digital health implementation literature. A brief synthesis in the discussion that explicitly frames these results as design and implementation lessons (e.g., implications for system speed, offline functionality, supervision models, and workflow integration) would enhance the manuscript’s relevance for digital health audiences.

Interpretive precision: In a few places, the narrative could be tightened to clearly distinguish between perceived improvements and objectively measured service outcomes. Minor wording adjustments would address this without changing the substance of the findings.

Minor editorial refinements- Address small grammatical issues and ensure consistent terminology (e.g., use of “mean” vs. “average”) would improve readability.

Reviewer #3: The manuscript is well written with great strengths such as the qualitative method used and the multicenter study conducted as well. However, since the journal is related to digital health, it would have been prudent to disclose the hardware specifications and the software architecture. Authors should provide a brief section on transition strategies for the increased workload and the inclusion of stakeholders, such as the health Ministry, since the cost implications would have to be considered in the national budget. There was also no explanation on why some participants did not receive text messages, any reasons for that, or whether this would affect a nationwide rollout.

**Do you want your identity to be public for this peer review?** For information about this choice, including consent withdrawal, please see our Privacy Policy

Reviewer #1: No

Reviewer #2: No

Reviewer #3: No

**Figure resubmission:**

**Reproducibility:** To enhance the reproducibility of your results, we recommend that authors of applicable studies deposit laboratory protocols in protocols.io, where a protocol can be assigned its own identifier (DOI) such that it can be cited independently in the future. Additionally, PLOS ONE offers an option to publish peer-reviewed clinical study protocols. Read more information on sharing protocols at https://plos.org/protocols?utm_medium=editorial-email&utm_source=authorletters&utm_campaign=protocols

---

## [Editor Report · Decision Letter 1]

6 Feb 2026

Pregnant women’s and health workers’ perceptions and experiences on the Rwandan ANC digital module intervention at selected health centres

PDIG-D-25-01089R1

Dear Dr Habtu,

We are pleased to inform you that your manuscript 'Pregnant women’s and health workers’ perceptions and experiences on the Rwandan ANC digital module intervention at selected health centres' has been provisionally accepted for publication in PLOS Digital Health.

Best regards,

Gloria Aidoo-Frimpong, PhD, MA,MPH

Academic Editor

PLOS Digital Health